# Tolerance to Stimulant Medication for Attention Deficit Hyperactivity Disorder: Literature Review and Case Report

**DOI:** 10.3390/brainsci12080959

**Published:** 2022-07-22

**Authors:** Kenneth Handelman, Fernando Sumiya

**Affiliations:** 1Centre for Integrative Mental Health, Assistant Clinical Professor of Psychiatry, McMaster University, Hamilton, ON L8N 3K7, Canada; 2ADHD Fellowship, Department of Psychiatry and Behavioural Neurosciences, McMaster University, Hamilton, ON L8S 3L8, Canada; fmsumiya@gmail.com

**Keywords:** stimulant medication, ADHD, tolerance, methylphenidate, amphetamines

## Abstract

Recommended treatment for attention deficit hyperactivity disorder (ADHD) includes stimulant medication. While these medicines are effective for most ADHD patients, benefits may wear off, suggesting tolerance. This paper reviews the published literature on tolerance to stimulant medication treatment for ADHD. As there are relatively few studies published, pivotal studies and ADHD treatment guidelines were also reviewed. Research demonstrates physiological changes related to continued stimulant usage in neurons and certain brain regions, suggesting a mechanism for tolerance development. One clinical study showed that 24.7% of patients developed tolerance to stimulants in the time of days to weeks; another showed 2.7% developed tolerance over 10 years. Long term follow-up studies demonstrate that medication response may lessen over longer durations of treatment in a high percentage of patients. Strategies to manage tolerance include switching stimulant medicines, drug holidays, or clinical reassessment. Three cases illustrate challenges with treating patients who develop tolerance to stimulant medication. The paucity of research and lack of guidance to clinicians may contribute to significant under recognition of tolerance to stimulant medication. Further research is required to define clinical tolerance for stimulants in ADHD and to provide guidance on identifying and managing tolerance in clinical practice.

## 1. Introduction

Attention deficit hyperactivity disorder (ADHD) is a neurodevelopmental disorder with symptoms of hyperactivity, impulsivity, and/or inattention. The symptoms affect cognitive, academic, behavioral, emotional, and social functioning [1]. A meta-analysis of 175 research studies worldwide on ADHD prevalence in children aged 18 and under found an overall pooled estimate of 7.2% [2]. The global prevalence of persistent adult ADHD was 2.58% and that of symptomatic adult ADHD was 6.76%, translating to 139.84 million and 366.33 million affected adults in 2020 globally [3].

ADHD has been linked to significant decreases in quality of life and functioning. In children and adolescents with ADHD, there are higher risks of school failure, parental and family conflict, social rejection by peers, low self-esteem, and delinquent behaviour [4]. In adolescence and adulthood, ADHD has been tied to academic and vocational underachievement, reduced occupational functioning, obesity, emotional dysregulation, unemployment, and suicide attempts [4]. Treatments for ADHD can reduce functional impairments and improve quality of life [4].

Stimulant medications are the most used and considered the most effective medications for treating ADHD children, adolescents and adults with ADHD, and they are recommended as first line medication in clinical treatment guidelines for ADHD [5,6,7,8]. Although stimulants can work well for most people with ADHD, clinicians may notice that some patients have a good initial response to stimulant medication, but then the benefits are lessened and tolerance may develop [9]. Tolerance is defined as the phenomenon whereby the effect of a drug decreases following repeated administration, requiring an increase in the dose to maintain the original efficacy [10]. The research on tolerance to stimulant medication is limited, and there are no clear recommendations in ADHD treatment guidelines for managing tolerance in clinical practice [5,6,7,8]. A literature review was performed to review the currently published research on tolerance to stimulant medication. We report on three retrospective case studies to share clinical aspects of tolerance to stimulant medications.

## 2. Materials and Methods

A literature review was performed in OVID Medline from 1946 to 2022. Search terms included: attention deficit hyperactivity disorder, methylphenidate, amphetamine, drug tolerance, tolerance. Then, an extensive search of references was performed on the papers found. Pivotal studies (such as the multimodal ADHD treatment (MTA) study and its longitudinal follow-up papers) and ADHD treatment guidelines were reviewed. Inclusion criteria for this review included: child, adolescent, or adult ADHD; English language; basic science research on the physiological impact of stimulant medications on the brain and/or neurons; specific research on tolerance to stimulant medications or related papers; treatment guidelines for the treatment of ADHD. This review yielded 23 papers. Three retrospective case reports describe the clinical management of tolerance to stimulants. These clinical cases were patients who developed early tolerance to stimulant medicine and who had complete or substantial loss of clinical benefit from stimulant medicine. They were drawn from a suburban specialty ADHD clinic that assesses and treats ADHD across the lifespan.

## 3. Results

The literature review will be divided into the sections of physiological studies on tolerance, clinical research, and a review of clinical ADHD treatment guidelines. Then, case reports will be shared.

### 3.1. Physiological Studies on Tolerance to Stimulant Medication

From a physiological perspective, chronic use of medication can cause brain changes. Animal models demonstrate that chronic treatment with methylphenidate (MPH) in rodents yields an attenuation of dopamine release probably due to an upregulation of the dopamine transporter or an increase in autoreceptor sensitivity [11].

Research also shows acute tachyphylaxis with MPH. Tachyphylaxis is defined as a rapid decrease in efficacy, often related to a rapid depletion of neurotransmitters [10]. Acute tachyphylaxis with MPH was demonstrated with PET scans, where intravenous MPH caused a fast adaptation in the brain to MPH [12]. Research in the dosing regimens in children with ADHD demonstrated that a flat pharmacokinetic dosing regimen of MPH lost 40% of its efficacy in the afternoon (compared with the same blood level in the morning), suggesting that acute tolerance occurred [13]. The researchers noted that although acute tolerance occurred, clinical observations found that tolerance does not develop over time. The researchers noted that the time course of acute tachyphylaxis suggests that acute tolerance would dissipate between the afternoon dose and the morning dose on the following day [13].

In a PET study of medication naïve adults with ADHD taking MPH for 12 months (compared with controls), it was shown that there was a significant increase (+24%, *p* < 0.01) in striatal dopamine transporter availability in the caudate, putamen, and ventral striatum [14]. Despite this measured change, clinical questionnaires showed that the response to ADHD medication was maintained throughout the year of treatment, though the authors speculate that the upregulation of dopamine transporters may decrease treatment efficacy and exacerbate symptoms while not under the effects of the medication [14].

These studies demonstrate pharmacodynamic tolerance, which is defined as adaptive changes in the organs, tissues, or networks that are affected by the interaction of the drug with the receptors responsible for its effects [10]. Even though there are demonstrated changes with repeated medicine use, these studies do not demonstrate that people taking stimulants for ADHD experience a clinical tolerance to the effects of the medicine.

### 3.2. Clinical Research on Tolerance to Stimulant Medication

A retrospective chart review of medication treatment of randomly selected ADHD patient documents that, of 166 patients treated with methylphenidate between 1976 and 1990, 68 (41%) required more than 60 mg of MPH per day, which is an off-label high dose of this medication. Of the 68 high-dose patients, 41 (60%) developed tolerance (they did not maintain a clinical response to the same dosage over time); that is 24.7% of the original sample (41/166 patients). The authors note that tolerance only developed in patients taking higher-dose MPH. Tolerance either developed within a few days or it could have taken more than a year. The authors noted that if patients developed tolerance to MPH they were switched to dextroamphetamine. If the substitute was less effective than MPH, the latter would be tried again in about one month, and in many cases, tolerance disappeared after one month [9].

A meta-analysis and a meta-regression were completed to assess the duration of treatment with pharmacological treatment for ADHD and efficacy [15]. They included 87 randomized controlled trials of treatment for ADHD, where the mean age was 21.9 years, the samples comprised mostly male patients, and there were moderately severe ADHD symptoms at baseline. Treatment duration was 9 weeks on average and ranged from 3–28 weeks. The bivariate model found that treatment duration was positively related with treatment efficacy. When other potential confounding variables were accounted for (baseline ADHD severity, type of drug, and comorbidity), only baseline ADHD severity was negatively associated with treatment efficacy [15]. This study did not find tolerance to medication treatment in studies with the treatment range of 3–28 weeks. The authors note that they excluded studies shorter than 3 weeks, as it is a minimum time to titrate medication. They acknowledge that while tolerance to medication could develop after 28 weeks, it was unlikely, as tolerance usually starts earlier [10].

In a study of longer-term response to methylphenidate treatment, researchers monitored the behavioural benefits of MPH in patients who were treated with medication from 3 years to 10 years. When the dose of MPH was adjusted for growth, the medication remained effective for the majority of the patients for 10 years. Only 3 of 108 patients (2.7%) lost the therapeutic response without an evident explanation other than the possibility of tolerance. The data showed that successful long-term treatment with MPH did not require increasing the dose beyond the need to adjust for body growth. The authors acknowledge that they only looked at children who were good responders to MPH, and if they had lost their response to MPH prior to 3 years of treatment, they would not have been included in this study [16].

In a study of 47 children with ADHD and reading disorder, researchers looked at different doses of MPH (0.3 mg/kg, 0.5 mg/kg, 0.7 mg/kg) twice daily for 6 months. They found that the 0.7 mg/kg group had significant improvements in teacher and parent rated hyperactive behaviours, as well as improved short-term memory and associative learning tasks. The teachers noted an increase in hyperactivity during the last three months of treatment, and this was not explained by MPH plasma levels. The cause of this change was suspected to be tolerance, though the authors acknowledge that it could possibly relate to situational issues, particularly because there was a (smaller) worsening in the placebo group as well in the last three months of the study [17].

In a meta-analysis and a meta-regression study of pharmacotherapy in adults with ADHD, researchers analyzed data from 44 studies with 9952 patients. The range of duration of the studies was 4–26 weeks. The analysis showed that the longer the study duration the smaller the efficacy of the pharmacological treatment for reducing ADHD symptoms. This may suggest chronic tolerance to the medication in adults treated for up to 26 weeks [18].

Two randomized withdrawal treatment studies have been reported, one with lisdexamfetamine (LDX) [19] and one with MPH [20]. In the LDX trial, after 26 weeks of LDX treatment, 157 children and adolescents were randomized to either continue LDX or take a placebo for 6 weeks in a randomized withdrawal period (RWP). During the RWP, significantly fewer LDX patients met failure criteria than the placebo. The study demonstrates the maintenance of efficacy of LDX in the treatment of children and adolescents with ADHD over 26 weeks. The rapid return of symptoms upon discontinuing LDX demonstrates the need for ongoing treatment [19]. In the MPH trial, 94 children and adolescents with ADHD who had been treated with MPH for 2 years were assigned to double-blind continuation of treatment for 7 weeks, or discontinuation of treatment for 7 weeks (with 3 weeks of taper of MPH and 4 weeks of placebo treatment). On average, the ADHD scores deteriorated significantly more in the discontinuation group than the continuation group. The researchers found that MPH treatment was still effective at 2 years, and discontinuing treatment led to worsened symptom ratings by both investigator and teacher-rated ADHD symptom ratings. That said, there were some participants who were able to stop the medication and not experience worsened symptoms. The authors note that many participants who were approached to participate in the study declined, and they speculate that this may have related to the fact that these patients/parents knew that the MPH was still needed (based on their experience of worsened symptoms with missed doses); in other words, the sample may have included patients who had milder ADHD symptoms or patients for whom their ADHD was resolving [20]. Taken together, these studies demonstrate that for the majority of patients stimulant medication is still effective after 6 or 24 months, and stopping medication generally worsens ADHD symptoms.

In the pivotal National Institutes of Mental Health funded multimodal treatment study (MTA), 579 children with combined-type ADHD were treated for 14 months in a randomized study design that compared medication management, behavioural therapy, combination treatment, and community control [21]. There was naturalistic follow-up for 16 years, with a local normative comparison group recruited at 24 months and followed for the subsequent 14 years. At the 36-month follow-up evaluation of the patients, growth mixture modeling found three latent classes. Class 1, which comprised 34% of the children in the study, had the smallest initial benefit to medication in the study, but their medication effects increased over time and were significant at the 36-month assessment. In Classes 2 and 3, the medication benefits were larger than Class 1 at 14 months; however, by the 36-month assessment, the medication benefits were not significant. This demonstrated that for the majority of children treated with medication (66%), beyond the 24-month assessment point in the MTA protocol, the overall effect of medication treatment was not beneficial for the reduction of ADHD symptoms. This suggests the possibility of waning benefit for continued medicine beyond 2 years for children with combined type ADHD [21]. In a review paper summarizing the long term outcomes from the MTA study, Hinshaw et al., write: “medications targeting dopamine and norepinephrine neurotransmission may, at least in some cases, have an ‘expiration date’ with respect to their effectiveness over the course of continuous administration… In short, just like behavioral treatments, which are known to lack continued benefit once the contingencies are lifted, medication for ADHD may not always be a viable lifelong option” [22].

Sibley et al., analyzed the course of ADHD during the naturalistic follow-up of the MTA study, from each of the assessments from 2–16 years. They found that approximately 30% of children with ADHD experienced full remission at some point during the follow-up period; however, a majority of them (60%) experienced recurrence of ADHD after the initial period of remission. Only 9.1% of the sample demonstrated recovery by study endpoint, and only 10.8% demonstrated stable ADHD persistence across the study time periods. Overall, 63.8% of participants with ADHD had fluctuating periods of remission and recurrence over time [23]. The natural course of ADHD over many years may impact a clinician’s assessment of the effectiveness of medication and whether tolerance has developed.

Two research papers addressing clinical treatment of ADHD were relevant, one on ADHD drug holidays [24] and the other on a practical evidence-informed approach to managing stimulant refractory ADHD [25]. In the drug holiday paper, researchers reviewed the literature and identified 22 studies published that documented research on drug holidays. They found that drug holidays are prevalent in 25% to 70% of families with children taking stimulant medication for ADHD and are more likely to be taken during school holidays. The reason for the drug holidays were to see whether medicine was still needed and to manage side effects (such as growth slowing) and for drug tolerance. One of the studies documented that doctors used breaks from medication to allow the body to readjust to the stimulant and to avoid the need to raise the dose of the medication; in other words, to reset tolerance to the medicine [24]. In the study that addressed the treatment of ADHD that is refractory to stimulant medication, the authors reviewed the treatment guidelines, used their expert clinical knowledge, and reviewed the published literature. They defined refractory ADHD as a failure to remit, minimal improvement, partial response with persistence of impairments, or no benefit at all to medication. They note that, to deal with refractory ADHD, it is important to: optimize stimulants for ADHD core symptoms; try alternative monotherapies for ADHD core symptoms; try non-stimulants for ADHD; use combination pharmacotherapy; use off-label medications with evidence that they help ADHD; treat comorbid conditions with ADHD. They note that some patients may develop tolerance to stimulant medication, but the extent and frequency of this is not understood. They note that raising the dose of stimulant medication may provide a temporary solution, but a short drug holiday may help with tolerance [25].

These studies are summarized in Table 1.

### 3.3. ADHD Treatment Guidelines and Tolerance

Published ADHD treatment guidelines were reviewed, looking for direction to clinicians around tolerance to stimulant medication. The following ADHD guidelines were reviewed: American (child and adolescent ADHD) [6], European (adult ADHD) [7], Canadian (child, adolescent, and adult ADHD) [5], and German (child, adolescent, and adult ADHD) [8]. Each guideline discusses the importance and usage of stimulant medication for ADHD and includes comments about what to do if the medicine is not effective. Only the Canadian ADHD treatment guidelines comment specifically about tolerance [5]. They note that some patients may confuse the energetic, mood, or pleasure side effects of a stimulant with the attention and behaviour control clinical effects. Even when “the energetic side effect tends to be reduced over time, the improvement of sustained attention is usually still there” [5]. They further note that if there are escalating doses of stimulants, or other atypical responses to them, clinicians should reevaluate the treatment goals or the appropriateness of that treatment for the individual [5].

### 3.4. Case 1: Patient A

A was seen in consultation in April 2020 at the age of 38 years old. She had been diagnosed with ADHD as a child and briefly took MPH immediate release but did not want to continue it. She restarted ADHD medication in the recent past and was referred for assessment. She was taking LDX 40 mg daily at the time of consultation. Her diagnoses were: ADHD combined presentation, binge eating disorder, cannabis use disorder, past history of substance use disorders (cocaine and ecstasy) in remission for 10 years, polycystic ovarian syndrome, and prediabetes. When she started LDX, it was helpful, but the benefits were wearing off early in the day after several weeks. At the initial consultation, the dose of LDX was increased by adding a midday dose of LDX 10 mg. The patient also started cognitive behaviour therapy (CBT) for adult ADHD. A pattern developed that A would have benefit to increased doses of LDX, but the benefits disappeared after days or weeks, and the medicine would wear off earlier and earlier in the day. With discussions of the safety of “off-label” dosing, and with blood pressure monitoring, A was doing better with LDX 110 mg daily (with LDX 70 mg in the morning, 20 mg at 11 a.m., and 20 mg mid-afternoon). After several months, the LDX stopped working completely.

A was switched to OROS MPH, with a rapid upward titration of dosing. She reached a dose of OROS MPH 117 mg daily, with OROS MPH 72 mg in the morning and 45 mg midday. This proved somewhat helpful, though not as helpful as the LDX. The benefits started to wear off, and after 4 months she was not experiencing benefit from the medication anymore. She was switched back to LDX.

While working with A over time, a pattern was established: she did better with LDX than she did with OROS MPH, it provided better symptom control overall for her ADHD and functioning. When she took OROS MPH, she wanted to minimize the time she took it for. Initially, she would take it for 4 weeks in an attempt to reset her tolerance to LDX, and it was successful. Since it did not work as well for her ADHD, we tried lowering the duration of OROS MPH as a break from LDX. When A tried it for only 7 days, the tolerance to LDX did not reset, and it was eventually discovered that she required at least 10 days of MPH to reset her tolerance to LDX. Furthermore, while initially A took LDX for around 2–3 months and waited for its benefits to drop off before taking MPH to reset the tolerance, it was decided to proactively switch the medicine earlier to avoid the loss of benefit from the LDX and to maintain more stability.

After approximately 1 year of adjusting and experimenting, A found that taking LDX for 5 weeks followed by MPH for 10 days provided the most stability for her and allows her to function best. Her final dosing is: LDX 110 mg daily (70 mg AM, 40 mg 2 h later); OROS MPH 117 mg od (72 mg AM, 45 mg 1 h later). She continues with CBT for ADHD, has lowered her cannabis use, and started working again after a long hiatus of unemployment.

### 3.5. Case 2: Patient B

B is a 17-year-old man who was diagnosed with ADHD combined presentation, mild oppositional defiant disorder, and adjustment disorder with anxiety. He had taken OROS MPH 72 mg daily and, while it worked well for him, it stopped working completely after 2 months. He tried other formulations of long-acting MPH with no success. He tried amphetamine-based medicines, including LDX and mixed amphetamine salts extended release (MAS XR). LDX did not provide symptom relief and contributed to anxiety symptoms. MAS XR worked for one week and then the benefits wore off.

B was tried on several other ADHD medications, including: guanfacine XR (GXR), up to 3 mg daily, in combination with stimulant medicines. He did not find that it helped at all, and no benefits were noted. He tried MPH Immediate release and, while it was helpful, the benefits disappeared after 2 weeks. He tried dextroamphetamine immediate release and, while it was helpful, the benefits wore off after 1 week. He tried atomoxetine (ATX) 60 mg daily (1 mg/kg/day) and there were no benefits plus significant nausea side effects. He tried bupropion XL 300 mg daily (as an off-label treatment for ADHD) and there were no benefits for focus, and he had significant insomnia. He tried modafinil 100 mg daily (as an off-label treatment for ADHD) and it did not help focus, but worsened insomnia.

B stopped all ADHD medication and also stopped his post-secondary education/training. His anxiety got worse, and he was treated with an SSRI, sertraline at 150 mg daily, and it was somewhat helpful for his anxiety symptoms, though his ADHD remained untreated.

### 3.6. Case 3: Patient C

C is a 10-year-old girl who was diagnosed with ADHD combined presentation by her pediatrician. She has no comorbid diagnoses. Her father has ADHD, both grandfathers have alcohol use disorders, and one also has bipolar disorder. There are no significant stressors in the family, apart from the recent global pandemic. She was initially prescribed LDX and it worked well for her symptoms. Within weeks, the medicine wore off earlier in the day, until it stopped working by noon. When it wore off, she experienced significant symptoms, including worsened focus, hyperactivity, anxiety, and emotional dysregulation. Immediate release dextroamphetamine (Dex IR) was added midday to extend the benefits of LDX with good effect. When this combination of medicines’ benefits wore off after several weeks, GXR was added and brought up to 3 mg daily. The GXR helped the stimulant medicines to last longer in the day and C did not experience significant negative symptoms when the medicine was wearing off.

Eventually, the benefits of LDX, Dex IR, and GXR were not significant, i.e., she had become tolerant. C was put onto OROS MPH and it worked well initially, but then the medicine was wearing off earlier in the day and the dose had to be increased. Eventually, she reached OROS MPH 72 mg daily, with MPH IR 7.5 mg midday, and she continued on GXR 3 mg daily. C did well with this for about 3 months and then the medicine was no longer helping.

She then switched back to LDX with GXR. When she first switched over to LDX, it lasted long enough and she did not need a midday dose of DEX IR. Then, the pattern repeated. She has been on LDX, Dex IR, and GXR for about 5 months and she has become tolerant; she will soon switch back to the MPH medications.

## 4. Discussion

ADHD is a condition that starts in childhood and lasts into adulthood for most patients. There are significant impairments related to ADHD. Standard treatment includes stimulant medication, among other possible medicines and psychotherapy. While stimulant medicines can work very well for ADHD, there is a concern about tolerance developing to these medications.

Physiological research demonstrates the biological adaptation to medication. With stimulant medication, there are changes in neurons and brain regions that can explain the mechanism of pharmacodynamic tolerance. The studies reviewed document that, although their studies demonstrated cellular changes, the stimulant medicines continued to work clinically.

Clinicians have observed patients who have lost benefits to stimulant medicine over time (i.e., the benefits “wear off”), though it can be hard to establish whether this is tolerance to the medicine, other clinical factors such as poor adherence to treatment, comorbid conditions, or the natural course of ADHD over time. The 16-year follow-up of the MTA study demonstrates that over time, ADHD may worsen or improve, related to the natural course of ADHD [23]. As such, a clinician’s assessment of medication effectiveness over time may be impacted by the natural worsening or improving of the disorder itself over months or years. This may lead to a misattribution of continued medication benefits or loss of medication benefits (i.e., tolerance), when it may actually relate to the natural waxing and waning of symptoms of ADHD over the course of the disorder.

A 2002 paper suggested that tolerance to stimulant medicine was common, occurring in 24.7% of randomly selected charts, and, interestingly, all the cases of tolerance were in patients who were on doses higher than 60 mg MPH per day [9]. In a study looking at response to stimulants over 3–10 years of treatment in children, the rate of tolerance was found to be 2.7% [16]. In the 36-month review of the MTA study, researchers showed that for 66% of the patients, the benefits of stimulant medication were notable at the start but wore off over time [21]. In a meta-analysis of 44 studies in adult ADHD medication treatment with 9952 patients, the studies had a range of 4–26 weeks. The longer the study duration the smaller the efficacy of pharmacological treatment for reducing ADHD symptoms [18]. These studies demonstrate that there can be a gradual loss of benefit from stimulant medicines over time with treatment.

While Ross et al., [9] discuss patients whose benefits wear off quickly from the stimulants, the MTA study [21], the meta-analysis of adult medication studies [18], and the 3–10-year follow-up of medication treatment [16] suggest that there can be slow and gradual loss of benefit to stimulant medication treatment.

There is no clear definition of tolerance to stimulant medication. These different studies suggest that clinicians should consider “early tolerance” for the clinical situation of losing the benefits of stimulant within days or weeks, and a more gradual or “late tolerance” for the clinical situation of losing the benefit of stimulant medicine over the course of months or years.

Furthermore, the extent of clinical benefit lost related to tolerance is not always documented clearly in the research. Some studies refer to complete loss of benefit to the medication, and other studies refer to a reduction of clinical benefit obtained from the medication. To assist with the clinical care of patients, future research could consider documenting whether there is “partial tolerance”, i.e., some reduction in ADHD symptom control, or “complete tolerance”, referring to a substantial or complete loss of benefits of the medicine.

Published ADHD treatment guidelines are helpful summaries of the research and its clinical application to treat patients with ADHD. Unfortunately, they do not provide guidance to clinicians around defining tolerance to stimulant medication, how to identify tolerance, nor how to clinically approach tolerance when it occurs.

When patients experience tolerance to stimulant medicines, the research suggests the following strategies: medication holidays to reset the tolerance [24,25]; switching between stimulant families, i.e., from MPH to amphetamines (AMPH) or vice versa [9]. Both the Canadian ADHD treatment guidelines and a paper on treatment refractory ADHD recommend that if patients lose their benefits to medicine or have a partial/inadequate response, the clinician should reevaluate treatment or address comorbid conditions which may be impacting treatment response [5,25].

Three clinical cases reviewed demonstrate patients who clinically develop tolerance to stimulant medicines in a short period of time, i.e., “early tolerance”. They also experienced “complete tolerance”. In two of the cases, clinical care has been successful by switching patients from one stimulant family (i.e., MPH) to the other (i.e., amphetamines AMPH), similar to the recommendation from Ross et al., [9]. Clinically, this has helped to “reset” the tolerance. In one of the cases, where patient B’s tolerance developed within days, i.e., 7–14 days, we were unable to find treatment for ADHD that proved helpful with any significant duration of effect, and he has untreated ADHD that has continued to cause impairment in his functioning.

Based on this review, the research suggests that there is a small percentage of patients with ADHD who develop “early tolerance” to stimulant medicines and a potentially larger percentage have a more gradual or “late tolerance” over years. Similarly, it seems that there are relatively few patients with ADHD who develop “complete tolerance” (complete loss of benefit of the medicine) and potentially a larger percentage who have “partial tolerance” (partial loss of benefit). Strategies to combat stimulant tolerance include: switching classes of stimulants (i.e., from MPH to AMPH and vice versa); taking medication holidays to reset the tolerance; using other treatments, such as psychotherapy, non-stimulant medications, and reassessing clinically (for factors such as medication adherence, comorbid conditions, or the natural course of ADHD over time).

There is a significant disparity between the reported rates of tolerance in the published literature (anywhere from 2.7% of patients over a 10-year study to 66% of children at 3 years to 9952 adults with ADHD losing some benefit of the medicine over a period of 26 weeks). There are demonstrated physiological mechanisms that underlie the biological basis of tolerance. Since there is a paucity of research on tolerance to stimulants, no clinical guidance in published ADHD treatment guidelines on identifying and managing tolerance to stimulant medication, and no clear definition of tolerance to stimulants, it is likely that tolerance to stimulant medicine is significantly under-recognized and under-reported. This is a significant clinical issue with a biological basis that urgently requires more research and clinical guidance. As the rate of stimulant usage is reported to have doubled in the United States of America between 2006 and 2016 [26], this issue takes on even more importance. Future research could elucidate: the incidence of tolerance (whether there is a difference in the rates of tolerance in youth compared with adults treated with stimulant medicine), provide a definition of tolerance which is relevant for clinicians treating ADHD patients, and provide more guidance on treatment approaches to address stimulant medication tolerance.

Furthermore, tolerance to psychopharmacology is not unique to ADHD. With mood disorders, studies document that patients treated with lithium for bipolar disorder may experience tolerance or discontinuation-induced refractoriness [27]. Treatment of major depressive disorder with antidepressant medicine can also lead to tolerance to medication in a significant percentage of patients [28]. While psychopharmacology is a powerful tool for treating psychiatric disorders, medication tolerance may decrease the effectiveness of medication treatment in the short term and/or longer term and may have a significant negative impact on patient outcomes.

## 5. Conclusions

Treatment of ADHD with stimulant medicine is generally effective and can help for many years. Research shows that some patients develop an “early tolerance” to these medicines, meaning they have an initial good response but the benefits wear off within days or weeks; some patients may develop more gradual or “late tolerance” to stimulants, where the benefits are lost over months or years of treatment; some patients also develop “complete tolerance” with a substantial or complete loss of clinical benefit to stimulants; some patients may develop “partial tolerance” with a partial loss of clinical benefit. There is insufficient research to clearly define clinical tolerance to stimulant medication in ADHD, and there are suggestions in the literature on strategies that may help, such as switching classes of stimulants (from MPH to AMPH and vice versa) to reset the tolerance or taking medication holidays and reassessing clinically for comorbid conditions or other clinical factors which may affect treatment response. There is a clear biological basis for stimulant medication tolerance, and the lack of sufficient research and guidelines may suppress recognition of this significant clinical issue and negatively impact patient outcomes. More research is needed and clinical guidelines should be updated to provide more guidance to clinicians on how to identify and manage tolerance to stimulant medication.

## Figures and Tables

**Table 1 brainsci-12-00959-t001:** Summary of Clinical Research on Tolerance to Stimulant Medicine.

Author	Title	Design	Main Findings
Ross et al. 2002 [9]	Treatment of ADHD when tolerance to methylphenidate develops	Retrospective chart review, *n* = 166	-**24.7% of patients developed tolerance**—some within days and some after one year of treatment -predictor of tolerance was needing higher/off label dosage of MPH -switched classes of stimulant medicine to treat tolerance
Castells et al., 2021 [15]	Relationship Between Treatment Duration and Efficacy of Pharmacological Treatment for ADHD: A Meta-Analysis and Meta-Regression of 87 Randomized Controlled Clinical Trials	Meta-Analysis and Meta-Regression of 87 randomized controlled trials; treatment duration was 3–28 weeks; 9 weeks on average; included children, teens and adults	-**This study did not find tolerance** to medication treatment in studies with the treatment range of 3–28 weeks. -Excluded studies shorter than 3 weeks
Safer et al., 1989 [16]	Absence of tolerance to the behavioral effects of methylphenidate in hyperactive and inattentive children	Retrospective chart review, *n* = 108; tracking stimulant treatment of ADHD for 3–10 years	-the dose of methylphenidate, when adjusted for growth, did not change significantly during the 3 to 10 years of treatment; -**Only 2.7% of patients lost the therapeutic benefit from medicine without an evident explanation other than the possibility of tolerance** -the dose calculations that minimized the effects of growth with age were milligrams per kilogram to the 0.7th power and milligrams per square meter of estimated body surface area
Kupietz et al., 1988 [17]	Effects of Methylphenidate Dosage in Hyperactive Reading-disabled Children: II. Reading Achievement	Prospective Study, *n* = 47; children with hyperactivity and reading disorder, treated for 6 months with methylphenidate immediate release twice daily	-Results showed positive effects of methylphenidate on reading that were mediated through behavioral control especially during the first 3 months of treatment. -The teachers noted an increase in hyperactivity during the last three months of treatment, and this was not explained by MPH plasma levels. -**The cause of this change was suspected to be tolerance, though the authors acknowledge that it could possibly relate to situational issues**, particularly because there was a (smaller) worsening in the placebo group as well in the last three months of the study
Cunill et al., 2016 [18]	Efficacy, safety and variability in pharmacotherapy for adults with attention deficit hyperactivity disorder: a meta-analysis and meta-regression in over 9000 patients	Systematic Review, meta-Analysis and Meta-Regression of 44 studies with 9952 adult ADHD patients; the duration of the studies was 4–26 weeks	-The analysis showed that the longer the study duration, the smaller the efficacy of the pharmacological treatment for reducing ADHD symptoms. -**This may suggest chronic tolerance to the medication in adults treated with stimulant medication for up to 26 weeks.**
Coghill et al., 2014 [19]	Maintenance of Efficacy of Lisdexamfetamine Dimesylate in Children and Adolescents With Attention-Deficit/Hyperactivity Disorder: Randomized-Withdrawal Study Design	Randomized Withdrawal Period (RWP) Study; *n* = 157 children and adolescents treated with lisdexamfetamine for 26 weeks underwent a 6 week randomized withdrawal period	-During the RWP, significantly fewer LDX patients met failure criteria than placebo -demonstrates the maintenace of efficacy of LDX in the treatment of children and adolescents with ADHD, and the rapid return of symptoms upon discontinuing LDX -**the study demonstrate that for the majority of patients, stimulant medication is still effective after 6 months, and stopping medication generally worsens ADHD symptoms.**
Matthijssen et al., 2019 [20]	Continued Benefits of Methylphenidate in ADHD After 2 Years in Clinical Practice: A Randomized Placebo-Controlled Discontinuation Study	Randomized Withdrawal Period (RWP); *n* = 94 children and adolescents who had been treated with methylphenidate for 2 years; assigned to double blind continuation or withdrawal of treatment over 7 weeks	-On average, the ADHD scores deteriorated significantly more in the discontinuation group than the continuation group. -The researchers found that MPH treatment was still effective at 2 years, and discontinuing treatment led to worsened symptom ratings by both investigator and teacher rated ADHD symptom ratings. -There were some participants who were able to stop the medication and did not experience worsened symptoms. -**The study demonstrate that for the majority of patients, stimulant medication is still effective after 24 months, and stopping medication generally worsens ADHD symptoms.**
Swanson et al., 2007 [21]	Secondary Evaluations of MTA 36-Month Outcomes: Propensity Score and Growth Mixture Model Analyses	Naturalistic follow-up of the NIMH Multimodal Treatment Study (MTA) at 36 months	-At the 36 month follow-up evaluation of the patients, growth mixture modeling found 3 latent classes. -In class 1, which comprised 34% of the children in the study, they had the smallest initial benefit to medication in the study, but their medication effects increased over time and were significant at the 36 month assessment. -In classes 2 and 3, the medication benefits were larger than class 1 at 14 months; however, by the 36 month assessment, the medication benefits were not significant. -**For the majority of children treated with medication (66%), beyond the 24 month assessment point in the MTA protocol, the overall effect of medication treatment was not beneficial for the reduction of ADHD symptoms.** -This suggests the possibility of waning benefit for continued medicine beyond 2 years for children with combined type ADHD
Sibley et al., 2022 [23]	Variable Patterns of Remission From ADHD in the Multimodal Treatment Study of ADHD	Analysis of the 16 year naturalistic follow-up of the NIMH Multimodal Treatment Study (MTA), reviewing the ADHD assessments from years 2–16	-Approximately 30% of children with ADHD experienced full remission at some point during the follow-up period; but a majority of them (60%) experienced recurrence of ADHD after the initial period of remission. -Most participants with ADHD (63.8%) had fluctuating periods of remission and recurrence over time; According to our review, theoretically, the natural course of ADHD may impact clinician’s ability to assess medication response during longer term follow up of ADHD medication treatment
Ibrahim et al., 2015 [24]	Drug Holidays From ADHD Medication: International Experience Over the Past Four Decades	Review of literature	-Drug holidays are prevalent in 25% to 70% of families with children/teens taking stimulant medication and are more likely to be exercised during school holidays. -They test whether medication is still needed and are also considered for managing medication side effects and drug tolerance. -One of the reviewed studies documented that doctors used breaks from medication to reset tolerance to the medicine
Cortese et al., 2021 [25]	Evidence-informed Approach to Managing Stimulant-Refractory Attention Deficit Hyperactivity Disorder (ADHD)	Review of literature, review of clinical guidelines, knowledge of expert practice in the field	-Refractory ADHD is defined as a failure to remit, minimal improvement, partial response with persistence of impairments, or no benefit at all to medication. -They note that to deal with refractory ADHD, it is important to: -Optimize stimulants for ADHD core symptoms; -Try alternative monotherapies for ADHD core symptoms; -Try non-stimulants for ADHD; -Use combination pharmacotherapy; use off-label medications with evidence that they help ADHD; -Treat comorbid conditions with ADHD. -Some patients may develop tolerance to stimulant medication, but the extent and frequency of this is not understood. -Raising the dose of stimulant may provide a temporary solution, but a short drug holiday may help with tolerance

Note: Main findings of the rate of tolerance have been bolded in the fourth column.

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
