# Peer review of "Tolerance to Stimulant Medication for Attention Deficit Hyperactivity Disorder: Literature Review and Case Report"

_brainsci, 2022, doi:10.3390/brainsci12080959_

Round 1

Reviewer 1 Report

The manuscript introduced tolerance to stimulant medication for ADHD. ADHD children, adolescents with ADHD and adults with ADHD are recommended as first line medication in clinical treatment guidelines for ADHD. The authors showed true state of tolerances using medicinelisdexamfetamine (LDX), methylphenidate (MPH), and guanfacine XR (GXR) ADHD patients. Stopping medication generally worsened ADHD symptoms. Thus, these findings will be useful for the treatment of ADHD, such as such as switching classes of stimulants (from MPH to AMPH and vice versa) to reset the tolerance. Therefore, the manuscript is not too excellent to be published. In other words, the manuscript is so excellent that it should be published.

Comments

(1) What differences were in tolerance results between children, adolescents and adults with ADHD?

(2) How were classes of stimulants used?

(3) Was it likely that tolerance results depended on patients?  What tendency was simply recognized?

(4) In line of 81, “doesn’t” should be replaced with “does not”.

(5) In line of 244 and 268, “didn’t” should be replaced with “did not”.

That is all.

Author Response

Point 1: the research as reviewed does not give clear answers about whether there are differences in tolerance to stimulant medicines based on stimulant treatment in children/teens compared to adults. Based on this insightful comment, I have added a sentence in the discussion to address this point.

Point 2: In the Ross et al paper - stimulant classes were switched if patients developed tolerance to one stimulant (ie methylphenidate) and then they were switch to the other family of stimulant (ie amphetamine) - or vice versa. In some of the published literature (ie longer term follow up studies), they didn't always discuss switching medicine ie in the 3-10 year follow up (Safer et al), they just discussed the ongoing benefit of methylphenidate medication (when adjusted for growth), and did not discuss switching to the other stimulants ie amphetamines. 

Point 3: the only patient characteristic that helped to predict tolerance to stimulants was in the Ross et al paper - where they indicated that it was only patients who required higher dose MPH that developed tolerance. That paper also reported the highest level of tolerance (almost 25%), which was unique/different than most of the other research. Further research will hopefully allow us to understand patient factors which put them at risk for tolerance to stimulant medication.

Points 4, 5: these have been fixed. 

Thank you for your review and comments. 

Reviewer 2 Report

Tolerance is defined as the phenomenon whereby the effect of a drug decreases following repeated administration, requiring an increase in the dose to maintain the original efficacy. The research on tolerance to stimulant medication is limited, and, to date, there are no clear recommendations in ADHD treatment guidelines for managing tolerance in clinical practice. In the present study the Authors conducted a literature review to evaluate the currently published information on tolerance to stimulant medication. They also report on three retrospective case studies to share clinical aspects of tolerance to stimulant medications. 

Overall, I believe that this paper is very interesting, well written, timely and scientifically sound. I have only some minor suggestions aimed to improve the quality of the paper and these are outlined below:

1) The tolerance can be seen also in several medications used for example in mood disorders. Please, add a brief comment on this point with appropriate references (see dois 10.1016/j.phrs.2018.10.025 and 10.2174/1745017901612010142).

2) Why the Authors didn’t conducted a systematic review than a narrative one? Please explain.

3) A table with most relevant paper should be added as it might be interesting and very useful to the reader.

Author Response

Point 1: I have added points about tolerance to Lithium and Antidepressant medicines in the discussion. I have added the excellent reference you suggested about antidepressant tolerance, and chose a different one for lithium tolerance (as I found it to be more clear on the points). Thank you for this insightful suggestion. 

Point 2: We chose to do a narrative review (rather than a systematic review) related to the fact that in our assessment, there is insufficient research on tolerance to stimulant medication to review systematically to develop a clearer understanding. By using a narrative review, we were able to explore other research papers which would not have been found in a systematic review, but contribute to the understanding and discussion of this topic - ie reading through the long term follow up of the NIMH Multimodal treatment study (to find papers at the 3 year follow up and 16 year follow up which contribute to the understanding of this topic); the papers on drug holidays and stimulant medication refractory ADHD just touch on tolerance, but contribute to this topic; and also reviewing published ADHD treatment guidelines is helpful as well. 

Point 3: We have created a table of the "Clinical Research on Tolerance to Stimulant Medicine" as you suggested. I am uploading the excel spreadsheet here; and I ask that the journal staff help to format it appropriately for the manuscript. Thank you for the suggestion.